# Impact of Irradiation on the Adhesive Performance of Resin-Based Dental Biomaterials: A Systematic Review of Laboratory Studies

**DOI:** 10.3390/ma16072580

**Published:** 2023-03-24

**Authors:** Florin Eggmann, Jonathan D. Hwang, Jose M. Ayub, Francis K. Mante

**Affiliations:** 1Department of Preventive and Restorative Sciences, Robert Schattner Center, Penn Dental Medicine, University of Pennsylvania, Philadelphia, PA 19104, USA; 2Department of Periodontology, Endodontology, and Cariology, University Center for Dental Medicine Basel UZB, University of Basel, CH-4058 Basel, Switzerland

**Keywords:** dental materials, dental bonding, permanent dental restoration, head and neck neoplasms, radiotherapy, radiation oncology

## Abstract

Head and neck cancers are a significant global health burden, with radiation therapy being a frequently utilized treatment. The aim of this systematic review was to provide a critical appraisal of laboratory studies that assessed the effect of irradiation on the adhesive performance of resin-based biomaterials. The analysis included 23 laboratory studies obtained from five databases, with most studies using human enamel, dentin, or both, and bonding procedures involving the fabrication of direct restorations, standardized specimens, bonding of orthodontic brackets, and luting of endodontic fiber posts. The protocols used for irradiation varied, with most studies exposing specimens made from extracted teeth to irradiation using cabinet irradiators to simulate treatment of head and neck cancer. The findings indicate that irradiation reduces the bond strength of dental adhesives and resin-based composites on flat, ground enamel and dentin specimens, with different adhesives and timing of irradiation having a significant impact on adhesive performance. Irradiation also increased microleakage in most studies. The effect of irradiation on marginal adaptation of direct resin-based composite restorations was inconclusive. This systematic review indicates that irradiation has detrimental effects on the adhesive performance of resin-based biomaterials and highlights the need for further clinical and laboratory studies evaluating the performance of adhesive materials and approaches to improve it.

## 1. Introduction

Head and neck cancers pose a significant global health burden, accounting for approximately 5.7% of all cancer-related deaths worldwide [1,2]. Treatment options for head and neck cancers vary depending on the diagnosis and stage of the disease. Radiation therapy is a frequently utilized treatment for head and neck cancers, either as a standalone option or in conjunction with chemotherapy, surgical intervention, or both [3,4,5,6,7].

Between 30% and 40% of patients with head and neck cancer are diagnosed with stage I or II disease, which can often be effectively treated with either surgery alone or definitive radiation therapy alone [5]. Over 60% of head and neck squamous-cell cancer cases are diagnosed at stage III or IV, indicating large tumors with marked local invasion, regional node metastases, or both [5]. Treatment decisions in such cases depend on primary cancer size, location, disease stage, age, patient preferences, performance status, and coexisting conditions [5]. Surgical resection with elective neck dissection is preferred for oral cavity cancer, followed by adjuvant radiotherapy or chemoradiotherapy [5].

However, despite its efficacy, radiation therapy has significant toxicity with both acute and late side-effects. Acute side-effects of radiation therapy include mucositis, xerostomia, dysphagia, and taste disturbance or loss [8]. Late side-effects of radiation therapy can occur months or even years after treatment has ended. These can include radiation-induced fibrosis and radiation-related caries [8].

Radiation-related caries occur in approximately 29% of patients within 3 months of completing treatment [9,10]. This susceptibility to caries is largely due to changes in saliva quantity and quality, as well as direct radiation effects on enamel and dentin [11,12,13,14]. In addition, oral trismus and mucositis, common side-effects of radiation therapy, can lead to inadequate biofilm control and increased consumption of carbohydrate-rich foods, both contributing factors to radiation-related caries [8,10,15].

Patients with head and neck cancer who undergo radiation therapy require comprehensive dental care, with special measures needed for caries prevention and control [11,15,16,17]. Owing to the high incidence of radiation-related caries, restorative interventions are often necessary [10]. In recent years, there has been an increase in studies evaluating the impact of irradiation on enamel, dentin, and the adhesive performance dental biomaterials [12,18].

The adhesive performance of dental biomaterials refers to their ability to bond to tooth structure or other dental materials, creating a strong and durable bond. This performance is a critical factor in the success of many dental treatments, including orthodontic procedures and restoration of teeth with defects caused by caries, tooth wear, or dental injuries [19,20]. Dental biomaterials that exhibit excellent adhesive performance typically have a high bond strength, good retention, low microleakage, and minimal marginal gaps [20,21].

The purpose of this systematic review is to provide a critical appraisal of laboratory studies that assessed the effect of irradiation, performed to simulate head and neck cancer treatment, on the adhesive performance of resin-based biomaterials. By synthesizing the findings of these studies, this review aims to contribute to a better understanding of the impact of radiation therapy on dental biomaterials and to inform clinical decision making for the management of radiation-related dental complications.

## 2. Materials and Methods

### 2.1. Research Question

The protocol for this systematic review was prospectively registered in the International Prospective Register of Systematic Reviews (PROSPERO 2022, CRD42022384753). The systematic review, taking account of the Preferred Reporting Items for Systematic Reviews and Meta-Analyses (PRISMA) statement, addressed the following question using the PICO (population, intervention, comparison, and outcome) framework [22]: In specimens made from or featuring enamel and/or dentin (human, bovine), how does irradiation simulating head and neck cancer treatment performed ahead of bonding or after bonding affect the adhesive performance of resin-based biomaterials compared with unirradiated controls?

### 2.2. Eligibility Criteria

Studies with full study reports were selected according to the inclusion and exclusion criteria given below. No time or language restrictions were applied.

### 2.3. Inclusion Criteria

In vitro/laboratory studyUse of pre-irradiation or post-irradiation bonding with a resin-based dental biomaterial (i.e., dental adhesives, resin-based composites, resin-based luting materials, resin-modified glass ionomer cements, compomers, resin-based sealants, and blocks made from resin-based composite for use in computer-aided design and computer-aided manufacturing)Data on adhesive performance in terms of bond strength, marginal discoloration, microleakage, marginal adaptation, debonding, or interfacial fracture toughnessSpecimens made from or featuring human or bovine enamel, dentin, or both

### 2.4. Exclusion Criteria

In silico studyClinical studyAnimal studyCase reportReview articleStudy assessing laser irradiation as surface pretreatmentStudy assessing irradiation as disinfection methodPosterAbstract-only paper

### 2.5. Search Strategy

Five databases, Cochrane Library, Embase, OpenGrey through DANS, PubMed, and Web of Science, were searched on 10 January 2023. The search strings, which were as similar as possible and tailored to the controlled vocabulary and syntax rules of each database, are included in the Appendix A.

### 2.6. Selection Process

After removal of duplicates through manual review (J.D.H.), two investigators (F.E. and J.D.H.) independently screened the titles and abstracts of articles retrieved through the electronic search against the eligibility criteria and selected articles considered potentially relevant for this systematic review. During the screening, author names and journals were unblinded. After retrieving the full articles of potentially relevant studies, three investigators (F.E., F.K.M., and J.D.H.) independently assessed each study report according to the eligibility criteria. Discrepant judgments regarding study eligibility were resolved by consultation with a fourth investigator (J.M.A.). Reasons for exclusion were recorded.

### 2.7. Data Collection

Three investigators (F.E., F.K.M. and J.D.H.) independently extracted qualitative and quantitative data of included studies into pilot-tested, structured spreadsheets. A fourth investigator (J.M.A.) made a final decision in case of incongruous assessments. No unpublished data were sought from corresponding authors or other sources. Data were extracted for details of dental specimens assessed (enamel, dentin, or both; human, bovine, or both), number of specimens, specimen fabrication, specimen shape, sample grouping, irradiation protocol, adhesives and resin-based materials used, test methods used to assess adhesive performance, and main findings.

### 2.8. Risk-of-Bias Assessment

The risk of bias of included studies was assessed independently by three investigators (F.E., F.K.M., and J.D.H.) using the RoBDEMAT tool [23]. An individual RoBDEMAT was completed for each laboratory study included in the systematic review. A fourth investigator (J.M.A.) resolved any inconsistent appraisals.

## 3. Results

### 3.1. Included Studies

Figure 1 shows the results of the study selection process, which led to the inclusion of 23 laboratory studies, whose year of publication ranged from 2001 to 2023. Data extracted from the reports of these studies are reported in detail in Table A1. During full-text assessment, two studies were excluded because their study reports were available as abstract-only papers but not as full-text articles [24,25]. One study was excluded because irradiation was used as a method of disinfection of specimens [26]. One study contained no data on adhesive performance and was, therefore, excluded [27].

### 3.2. Characteristics of Included Studies

#### 3.2.1. Dental Specimens

Most studies used human enamel, dentin, or both. One study used specimens made from bovine teeth [28]. Two studies used deciduous molars to fabricate specimens [29,30]. Extracted molars were used in most studies to provide the bonding substrate, with some studies furnishing no information whether the sample comprised permanent molars, deciduous molars, or both.

#### 3.2.2. Bonding

The bonding procedures in the included studies comprised fabrication of direct restorations (Class I, II, and V), fabrication of standardized specimens with resin-based composite bonded on flat, ground enamel and/or dentin, bonding of orthodontic brackets, and luting of endodontic quartz fiber posts.

#### 3.2.3. Irradiation Protocols

Two studies used extracted teeth that had been exposed to an irradiation dose of ≥50 Gy during radiation therapy prior to tooth removal [31,32]. Most studies used cabinet irradiators to expose specimens made from extracted teeth to high-energy X-ray radiation. A total dose level of 60 Gy, usually achieved through daily, fractional irradiation over 6 weeks, was most used in the included studies. To examine possible dose–effect relationships, two studies exposed different groups to different total dose levels, ranging from 10 Gy to 70 Gy [29,33]. The timepoint of irradiation differed across the included studies; the bonding procedures were performed before, during, or after irradiation of the specimens, with some studies subjecting different samples at different timepoints to irradiation in order to assess the impact of the timing on adhesive performance. Two studies included a comparison between samples restored immediately after irradiation or with a 6-month period in between [30,34]. One study evaluated the effect of covering restored and unrestored teeth with C-shaped shields made from 0.5 mm thick aluminum during irradiation [35].

#### 3.2.4. Resin-Based Biomaterials

The adhesives used in the included studies comprised etch-and-rinse adhesives (two-step, three-step, and four-step), self-etch adhesives (one-step and two-step), and universal adhesives that were applied in self-etch or etch-and-rinse mode. One study assessed a resin-modified glass ionomer material in addition to resin-based composite [36]. Apart from the study that bonded quartz fiber posts in root canals using dual-curing resin-based luting material, most studies used light-curing resin-based composites to create standardized restorations or cylinders for bond strength testing and other assessments of adhesive performance.

#### 3.2.5. Methods to Evaluate Adhesive Performance

The methods employed to assess the adhesive performance of resin-based biomaterials included shear bond strength tests, tensile and microtensile bond strength tests, pushout bond strength tests, microleakage assessments using dye penetration, analysis of marginal adaptation using X-ray microtomography, and marginal gap measurements.

#### 3.2.6. Effects of Irradiation on Bond Strength

In seven studies that assessed bond strength on flat, ground enamel and/or dentin specimens, irradiation reduced the bond strength of dental adhesives and resin-based composite [30,33,34,37,38,39,40]. In one study, irradiation reduced the bond strength on dentin but not enamel [29]. No significant difference between irradiated and unirradiated specimens was found in three studies [32,41,42]. Two studies found that irradiation impaired the bond strength of some adhesives but not others [31,43].

Different adhesives obtained significantly different bond strength in six studies [30,31,31,34,42,43]. The enamel bond strength of a universal adhesive was higher when applied in etch-and-rinse mode compared with self-etch mode [41].

Irradiation decreased the push-out bond strength of endodontic quartz fiber posts, with the most pronounced drop in bond strength occurring in specimens subjected to irradiation prior to the adhesive luting procedure [44].

A decrease in enamel bond strength of orthodontic brackets was observed in three of four studies that compared irradiated enamel specimens with unirradiated specimens [45,46,47]. No significant decline in enamel bond strength of orthodontic brackets was found in one study [36].

The timing of irradiation in relation to the restorative procedure was found to have a significant impact on adhesive performance in all included studies that assessed this parameter [30,37,38,39,43,44]. According to three studies, the most pronounced detrimental impact was found in specimens subjected to irradiation before the restorative procedure [37,38,44]. One study found that the impact of the timing of irradiation differed depending on the type of adhesive used [43].

Both studies that evaluated dose–effect relationships reported a dose-dependent decrease in dentin bond strength [29,33]. In one of these studies, a similar exposure–effect relationship was also observed on enamel [33], while the other study did not find any such relationship [29].

#### 3.2.7. Effects of Irradiation on Microleakage

In three of four studies that evaluated microleakage, a higher degree of microleakage occurred in irradiated specimens than in unirradiated ones [28,35,48]. Lower degrees of microleakage were detected in teeth that had been covered with aluminum shields during irradiation [35]. No significant difference in microleakage between irradiated and unirradiated specimens was found in one study [49]. Specimens bonded with an etch-and-rinse adhesive showed more microleakage at the cervical dentin margin than specimens bonded with self-etch adhesives [49].

#### 3.2.8. Effects of Irradiation on Marginal Adaptation

The three included studies that assessed the marginal adaptation of direct resin-based composite restorations reported conflicting findings. Irradiation did not significantly affect the marginal adaptation of direct resin-based composite restorations bonded with a universal adhesive in one investigation [18]. In specimens subjected to irradiation before the restorative procedure, more marginal defects were observed in restorations bonded with a universal adhesive applied in self-etch mode compared with the same adhesive in etch-and-rinse mode [18]. One study found that the marginal gap of direct resin-based composite restorations was higher in irradiated than unirradiated specimens [40].

### 3.3. Risk of Bias

Table A2 reports in detail the results of the RoBDEMAT assessments of the included studies. The signaling questions of the RoBDEMAT tool are provided in the Appendix A. Randomization of samples, sample size rationale and reporting, and blinding of test operators were the domains where the RoBDEMAT assessments most frequently indicated insufficient reporting, methodological limitations, or both.

## 4. Discussion

This systematic review provided a critical appraisal of 23 laboratory studies assessing the effects of irradiation, performed to simulate head and neck cancer treatment, on bond strength, microleakage, and marginal adaptation of dental restorations made with resin-based biomaterials. Irradiation reduced the bond strength of dental adhesives and resin-based composites on flat, ground enamel and/or dentin specimens in most studies. Different adhesives and timing of irradiation had a significant impact on adhesive performance. Irradiation also increased microleakage in most studies. The effect of irradiation on marginal adaptation of direct resin-based composite restorations was inconclusive, with conflicting findings reported in the included studies.

Although this systematic review offers valuable insights into the effects of irradiation on adhesive performance of resin-based biomaterials, it is important to consider its limitations. The review was confined to laboratory studies that evaluated resin-based biomaterials, which limits the scope of its findings. Glass ionomer cements can release fluoride and act as refillable fluoride reservoir, potentially curbing the development of secondary caries, especially in high-risk patients undergoing radiation therapy [10]. However, it remains unclear which biomaterial is most suitable for treating radiation-related caries based on available clinical evidence [10]. This paucity of evidence underscores the need for further laboratory and clinical investigations that evaluate the performance of different biomaterials and explore methods for improving their effectiveness.

The risk-of-bias assessment of the included studies revealed some opportunities for improvement in methodological design and reporting, particularly in the areas of sample randomization, sample size justification and reporting, and blinding of investigators. To assist researchers in designing laboratory studies that assess dental materials and in drafting study reports, the RoBDEMAT checklist, along with reporting guidelines issued by the Equator Network (www.equator-network.org [accessed on 23 March 2023]), can be a useful tool [23].

Irradiation causes a degradation of the interprismatic substance of enamel [50]. There is also a decrease in crystallinity of hydroxyapatite and an increase in the protein-to mineral ratio [12]. As a result, achieving strong and stable bonding to irradiated enamel is more challenging compared to unirradiated enamel. Most of the studies on enamel bonding reported that irradiation reduces the bond strength. However, a few studies found no significant impact of irradiation on bond strength, including one that investigated the bonding of resin-modified glass ionomer cement [29,36,41]. The reduced bond strength observed in most studies is likely due to the microstructural changes in enamel caused by irradiation, which can make it more difficult for adhesives to bond effectively.

The effect of irradiation on dentin appears to be more severe than enamel owing to the higher content of water and organic matter. In dentin, decarboxylation caused by irradiation destroys the electrostatic linkages between the carboxylate and phosphate side-chains of collagen and leads to a decoupling of calcium from collagen side-chains [14,50]. These disruptions in collagen and obliteration of dentinal tubules may compromise the dentin bond by impairing the hybrid layer [30,41]. This effect is evidenced by the reduction in or loss of resin tags and thin hybrid layers reported for bonding to irradiated dentin [30]. However, one-half of the studies that investigated tensile bond strength to dentin reported that irradiation reduced bond strength, while the other half found no difference. For studies that investigated shear bond strength, a majority—seven out of nine studies—reported that irradiation led to a decrease in bond strength. However, one study using deciduous molars reported that a 6 month delay in bonding after irradiation improves the bond strength, suggesting that there may be some recovery of the damage over time [30].

In line with the body of published evidence, etch-and-rinse adhesives appear to provide higher bond strengths to both irradiated and control enamel than self-etch adhesives applied without prior phosphoric acid etching [21,33,51]. Interestingly, studies that investigated shear bond strength reported that self-etch adhesives provided higher bond strength to dentin [30,34]. In contrast, one study assessing microtensile bond strength reported that the tested etch-and-rinse adhesive achieved better bond strength than the self-etch adhesive on dentin [27]. These differences in findings may be due to variations in stress distribution at the test interface for the two modalities [52]. Additionally, consistent with strong evidence derived from clinical and laboratory studies, the findings of this systematic review indicate a significant difference in adhesive performance of different adhesives [20,21].

Assessments of microleakage can be used to evaluate the adhesive performance and quality of restorations. Four studies investigated microleakage of restorations placed in irradiated teeth. The majority of these studies reported higher microleakage in irradiated teeth compared with control teeth [28,35,48]. However, one study found no difference in leakage between irradiated and unirradiated teeth and reported better results with a self-etch adhesive than an etch-and-rinse adhesive at cervical margins of restorations [49]. Additionally, one study that evaluated marginal adaptation of irradiated teeth using X-ray microtomography found no difference in adaptation but more dentin margin defects in specimens bonded with a self-etch adhesive [18].

Conventional fractionation radiation therapy with a total dose level of 60 to 66 Gy is a standard of care in high-risk patients with squamous-cell carcinoma of the head and neck [4,5]. Most of the laboratory studies included in this review simulated this by subjecting specimens to a total dose of 60 Gy. However, the dosage of radiation that reaches tooth structure during radiation treatment can vary depending on tumor location and size, as well as the efficiency of targeting of radiation. Oral stents and lead shielding can be used to reduce radiation to surrounding tissues during radiation treatment [50]. Advances in the use of intensity-modulated radiation therapy are also expected to reduce damage to surrounding tissues [53]. One study evaluated the use of lead shielding to simulate the attenuation of radiation to teeth and reported that shielding reduced the microleakage of restorations [35]. As it is possible to place protective appliances on teeth or the entire dental arch, it is worth exploring approaches to minimize the harmful effects of radiation by utilizing such intraoral shielding devices. Further investigation in this area is warranted to determine the most effective methods for reducing radiation damage to dental structures.

According to the findings of this systematic review, patients with head and neck cancer who are scheduled to undergo radiation treatment should receive dental evaluation and necessary restorative treatments before radiation therapy begins. To minimize radiation exposure of teeth, measures such as lead shielding and stents should be used whenever possible. Given that bond strength of restorations that are placed 6 months after radiation may equal that of unirradiated teeth, restorations placed 6 months or longer after radiation therapy may obtain better adhesive performance compared with restorations placed within half a year of radiation therapy. While it is advantageous to use enamel conditioning with phosphoric acid, there is no consensus on the preferred adhesive strategy for bonding to irradiated dentin. Dental practitioners need to consider the difference in performance of different adhesives and select restorative materials with due care.

## 5. Conclusions

This systematic review generated several important conclusions that are relevant to both research and clinical patient care.

Evidence derived from laboratory studies suggests that irradiation has a detrimental effect on the adhesive performance of resin-based dental biomaterials.The long-term impact of radiation on dental adhesion remains unclear, but it is plausible that the adverse effects may lessen with time between radiation therapy and the restorative procedure. However, current evidence on this is scanty.Significant differences have been observed in the performance of different adhesives on irradiated enamel and dentin. To achieve favorable restorative outcomes, it is, therefore, crucial to choose adhesives with a proven performance record in both laboratory and clinical studies, and to take painstaking care during bonding and buildup procedures.Further research is necessary to gain a comprehensive understanding of the effects of irradiation on teeth with restorations, develop methods to mitigate the adverse effects of irradiation, and explore ways to improve the efficacy of dental restorations.

## Figures and Tables

**Figure 1 materials-16-02580-f001:**
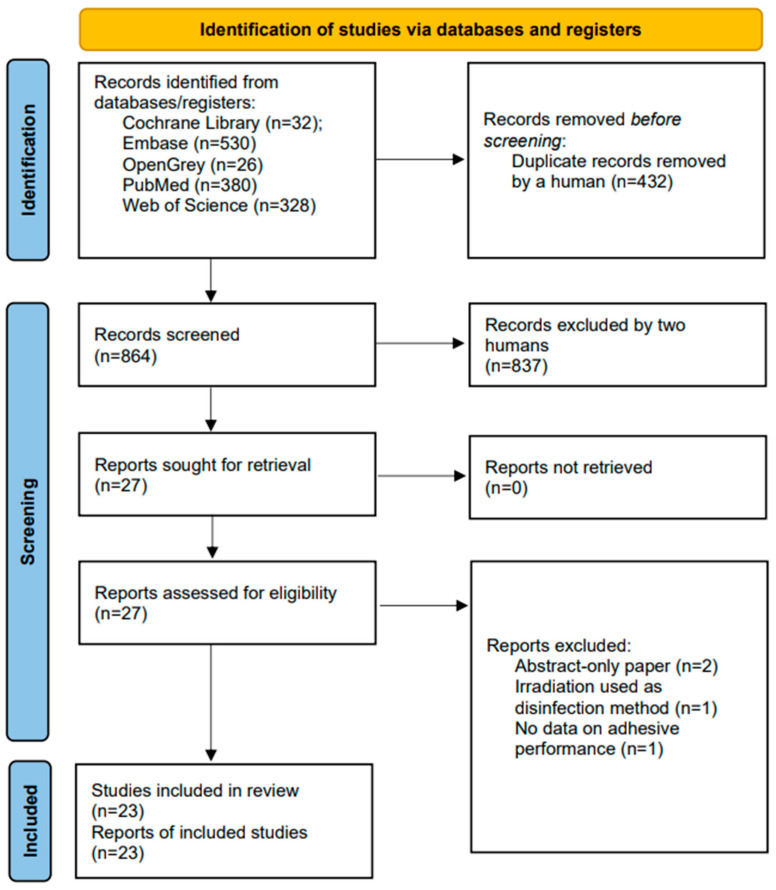
PRISMA 2020 flow diagram depicting the selection of records for this systematic review [22].

## Data Availability

This article and the Appendix A contain all data collected and analyzed in this systematic review. The BibTeX file of the electronic searches is openly available in Zenodo at https://doi.org/10.5281/zenodo.7662400.

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
