# Peer review of "Impact of Irradiation on the Adhesive Performance of Resin-Based Dental Biomaterials: A Systematic Review of Laboratory Studies"

_materials, 2023, doi:10.3390/ma16072580_

Round 1

Reviewer 1 Report

1.     No significant difference in irradiation and unirradiation in the specimens in the majority of study means, why to do irradiation?

2.     Inclusion criteria, Exclusion criteria, Search strategy, Selection process, Data collection, Risk of bias assessment in which way these subtitles are relevant to the title. Give your hypothesis.

3.     What type irradiations have been used?

4.     What kind of adhesive materials you have checked

5.     In introduction 3rd line give space between the words

6.     The content did not justify the title.

Reviewer 2 Report

The systematic review reveals that irradiation reduces adhesive performance and increases microleakage in resin-based biomaterials, emphasizing the need for improved adhesives. I think the content of this review is comprehensive and suitable for publication in Materials; however, there are only three minor points that need to be revised.

1. To enhance the text, use active voice, and clearly state the research question in the systematic review. Providing additional context on the prevalence of radiation therapy for head and neck cancers, types of side effects resulting from radiation therapy, or the meaning of "adhesive performance" of resin-based biomaterials can further clarify the article.

2. To make Table A1 more intuitive and easier to understand and compare, consider arranging the information according to the test method or the dental specimens.

3. The conclusion effectively summarizes the main points of the article, but it could be more helpful by providing guidance or advice to help readers make informed decisions.

Round 2

Reviewer 1 Report

References needs to be improved and if possble few more can be added.

Author Response

General comment

We appreciate your expeditious review of the revised report. Taking your valuable feedback into account, we have amended the report. You find the point-by-point replies to your suggestion below. The changes made to the report are highlighted in yellow.

We thank you for taking the time to review the revised study report and look forward to receiving your feedback soon.

Reply to reviewer #1

1. Concern of the reviewer

References needs to be improved and if possible, few more can be added.

Our response

Thanks for your constructive input. We have undertaken a meticulous review of the reference list to ensure that pertinent references are given. Additionally, based on valuable feedback, the amended report includes further references to high-level evidence in reputable journals to support factual statements and relate the findings of our systematic review to relevant investigations. We have been grateful for your positive feedback, which helped improve the overall quality of the report.

Revised text

See revised and amended reference list.